# Bacterial Species from Vaginal Microbiota Differently Affect the Production of the E6 and E7 Oncoproteins and of p53 and p-Rb Oncosuppressors in HPV16-Infected Cells

**DOI:** 10.3390/ijms24087173

**Published:** 2023-04-12

**Authors:** Sabrina Nicolò, Alberto Antonelli, Michele Tanturli, Ilaria Baccani, Chiara Bonaiuto, Giuseppe Castronovo, Gian Maria Rossolini, G. Mattiuz, M. G. Torcia

**Affiliations:** 1Department of Clinical and Experimental Medicine, University of Florence, 50134 Florence, Italy; 2Clinical Microbiology and Virology Unit, Careggi University Hospital, 50139 Florence, Italy

**Keywords:** HR-HPV (high-risk human papillomavirus), CIN-1 (low grade of cervical intraepithelial neoplasia), *Lactobacillus*, *Gardnerella*, *Megasphaera*, E6 and E7 oncoproteins, p53 and pRb production

## Abstract

Vaginal dysbiosis is characterized by a decrease in the relative abundance of *Lactobacillus* species in favor of other species. This condition facilitates infections by sexually transmitted pathogens including high risk (HR)-human papilloma viruses (HPVs) involved in the development of cervical cancer. Some vaginal dysbiosis bacteria contribute to the neoplastic progression by inducing chronic inflammation and directly activating molecular pathways involved in carcinogenesis. In this study, SiHa cells, an HPV-16-transformed epithelial cell line, were exposed to different representative vaginal microbial communities. The expression of the HPV oncogenes *E6* and *E7* and the production of relative oncoproteins was evaluated. The results showed that *Lactobacillus crispatus* and *Lactobacillus gasseri* modulated the basal expression of the *E6* and *E7* genes of SiHa cells and the production of the E6 and E7 oncoproteins. Vaginal dysbiosis bacteria had contrasting effects on *E6/E7* gene expression and protein production. The expression of the *E6* and *E7* genes and the production of the relative oncoproteins was increased by strains of *Gardnerella vaginalis* and, to a lesser extent, by *Megasphaera micronuciformis.* In contrast, *Prevotella bivia* decreased the expression of oncogenes and the production of the E7 protein. A decreased amount of p53 and pRb was found in the cultures of SiHa cells with *M. micronuciformis*, and accordingly, in the same cultures, a higher percentage of cells progressed to the S-phase of the cell cycle compared to the untreated or *Lactobacillus*-stimulated cultures. These data confirm that *L. crispatus* represents the most protective component of the vaginal microbiota against neoplastic progression of HR-HPV infected cells, while *M. micronuciformis* and, to a lesser extent, *G. vaginalis* may directly interfere in the oncogenic process, inducing or maintaining the production of viral oncoproteins.

## 1. Introduction

Cervical cancer represents the fourth most common cancer in women with 570,000 new cases diagnosed and an estimated 311,000 women died of the disease in 2018 [1]. Persistent infections with high risk-human papillomavirus (HR-HPV) strains represent a major risk factor for cervical cancer development. In particular, infections with HPVs-16 and -18 strains are responsible for around 70% of cases of cervical cancer worldwide [2]. Approximately 70% of HPV infections are cleared spontaneously in one year and 90% in two years. The infection persists in the remaining cases and can progress to low grade cervical intraepithelial neoplasia (LSIL/CIN-1), high grade squamous intraepithelial lesions (HSIL/CIN-2), or carcinoma *in situ* (CIN3). While the majority of CIN-1 is spontaneously cleared by the immune system, CIN-2 lesions progress to neoplasia in almost 50–60% of cases. The regression of CIN-2 is observed in almost 40% of women [3]. It is difficult to predict which lesions regress and which progress to carcinoma, and traditionally, women with a diagnosis of CIN-2 undergo surgical treatments that, fortunately, are now much less invasive than in the past [4].

Drivers governing transition between HPV acquisition, clearance, persistence, and cancer progression have not been elucidated yet. Cervico-vaginal microbiota in cooperation with the immune system maintains the health of the vaginal environment and largely contributes to prevent HPV infection as well as other sexually transmitted infections. In particular, *Lactobacillus crispatus*, *Lactobacillus gasseri*, and *Lactobacillus jensenii* were reported as the most protective species against HR-HPV infection and persistence [5,6,7,8]. In contrast, *Gardnerella vaginalis*, *Sneathia*, *Megasphaera*, and *Prevotella* were repeatedly found associated with HR-HPV persistence and high-grade lesions in infected women. Occasionally increased prevalence of *Lactobacillus iners* has been associated with squamous intra-epithelial lesions and progression to dysplasia/cancer [9].

The cervico-vaginal microbiota associated with the recovery of high-grade lesions (CIN-2) was also studied. Regression of CIN-2 is significantly less frequent in women with a non-lactobacillus dominated microbiota [10]. The presence of *Megasphaera*, *Prevotella*, *Gardnerella* (bacterial vaginosis-associated genus), *Sneathia*, and *Atopobium* species has been significantly associated with the persistence of CIN-2 lesions and cancer progression [10].

Despite the huge amount of data on the composition of vaginal microbiota associated with the risk of cancer development, the molecular mechanisms activated by bacterial cells involved in the neoplastic transformation of cervical epithelial cells were not elucidated.

In this study, we investigated the effects of representative taxa of vaginal microbial communities on relevant steps of the oncogenesis process triggered by HR-HPV strains.

The production of HPV oncogenic proteins E6 and E7 and the amount of p53 and pRb oncosuppressors were investigated in SiHa cells, an HPV-16 transformed epithelial cell line cultured with live bacterial cells of the most representative species of cervico-vaginal microbiota.

## 2. Results

### 2.1. Effects of Vaginal Bacteria on HPV-16 E6 and E7 Gene Expression and Protein Production by SiHa Cells

As a first step of analysis, we investigated whether vaginal bacteria differently affected the expression of the viral *E6* and *E7* oncogenes and the production of the relative proteins by cervical epithelial HR-HPV-transformed cells. SiHa cells, harboring few copies of integrated copies of HPV-16, were used as the experimental model of HR-HPV transformed cells and co-cultured with live bacterial cells for 6 h.

Figure 1 shows the results of *E6* and *E7* gene expression obtained through RT-PCRs with primers specific to the *E6* and *E7* genes of HPV16.

The expression of the *E6* gene (panel A) was significantly decreased in cultures with *L. crispatus* and *L. iners*. The mean value of *E6* mRNA expression was always above that of the untreated cultures in *G. vaginalis* and *M. micronuciformis* and *A. vaginae* stimulated cultures. *P. bivia* was the only species associated with vaginal dysbiosis that significantly decreased *E6* gene expression.

When we studied the expression of the *E7* gene, we found that all vaginal lactobacilli decreased the basal expression of the oncogene with *L. crispatus* and *L. gasseri* more efficiently compared to *L. iners. P. bivia* was, again, the only species of vaginal dysbiosis bacteria that significantly decreased the basal *E7* oncogene expression.

The *G. vaginalis* strain, *M. micronuciformis*, increased the basal expression of the *E7* gene.

We also investigated the production of viral oncoproteins in the SiHa cells cultured with live bacteria, as above reported.

Figure 2, panel A shows that the amount of E6 proteins was significantly lower than the untreated controls only in cultures with *L. crispatus*.

Figure 2 panel B shows that the production of the E7 protein was significantly decreased in cultures with *L. crispatus* and *L. gasseri.* Surprisingly, the E7 protein was found increased in cultures with *L. iners*. Taken in consideration with the results of the gene expression analysis, these data suggest that *L. iners* may increase the stability of the E7 oncoprotein.

Among the vaginal dysbiosis bacteria, E7 was significantly decreased in cultures with *P. bivia* while it was increased in cultures with *G. vaginalis* and *M. micronuciformis*. (*p* < 0.05 and *p* = 0.05, respectively).

Overall, these data indicate *L. crispatus* and *L. gasseri* as vaginal *Lactobacillus* species that are able to modulate the expression of *E6* and *E7* oncogenes and the production of the relative oncoproteins.

Among the vaginal dysbiosis bacteria, *P. bivia* was the only species that modulated *E6* and *E7* oncogene expression and the production of the relative oncoproteins.

*G. vaginalis* and *M. micronuciformis* did not affect the basal production of E6 and increased the production of the E7 oncoprotein.

### 2.2. Effects of Vaginal Bacteria on p53 and pRb Production by SiHa Cells

As a third step of our study, we explored the production of the tumor suppressor p53 and pRb proteins. Loss of function of these important proteins plays a significant role in most human cancers including cervical cancer. The HPV oncoproteins E6 and E7 target the ubiquitination of p53 and pRb proteins, respectively [11,12]. To investigate whether vaginal bacteria affect the production of p53 and pRb proteins, SiHa cells were cultured with live bacterial cells as reported and the production of p53 and of pRb was evaluated by Western blot analysis.

Figure 3 shows that the production of the p53 and pRb proteins was differently affected by co-culture with vaginal bacteria. While p53 and pRb production was unaffected or increased in culture with vaginal lactobacilli, the amount of the two oncosuppressors was significantly lower in cultures with *M. micronuciformis*, suggesting that this species might induce deregulation of the cell cycle.

### 2.3. Effects of Vaginal Bacteria on Cell Cycle

Loss of p53 and pRb as induced by the viral E6 and E7 oncoproteins allow for a high number of cells to enter in the S phase of cell cycle and even to progress to G2/M [13].

As a final step of the study, we evaluated the effects of vaginal bacteria on phases of the cell cycle. SiHa cells were cultured with live bacterial cells for 24 h and cell proliferation was recorded by FACS analysis using CSFE fluorescent staining.

Figure 4 shows that vaginal lactobacilli do not affect any phase of the cell cycle. The percentage of cells in G0, G1, S, G2/M were not different from those revealed in the untreated controls.

In contrast, a significant increase in the percentage of cells in the S phase of cell cycle compared to the untreated control was found in SiHa cultured with *M. micronuciformis*.

According to the results previously reported, *M. micronuciformis*, and to a lesser extent, *G. vaginalis*, are strongly suggested as bacterial species more involved in affecting normal cellular functions.

## 3. Discussion

Vaginal microbial communities dominated by *Lactobacillus* species strongly contribute to protect the vaginal environment from infections with sexually transmitted pathogens including HR-HPV. Infection with high-risk HPV strains is cleared by the host immune response in most cases. In a small percentage of women, however, the infection persists, and in some cases, women develop high-grade cervical intraepithelial neoplasia [14].

Either longitudinal or cross-sectional studies have revealed that *Lactobacillus*-dominated microbiota is highly protective against HR-HPV infection. The role of vaginal lactobacilli in HPV disease, however, is not limited to preventing viral infections.

The clearance of HPV infection requires the coordinate activation of both innate and adaptive immunity and the differentiation of T helper effector lineages and cytotoxic T lymphocytes (CTLs) are needed for the clearance of HPV [15]. HPVs, and HR-HPV in particular, have evolved sophisticated mechanisms of immune escape, resulting in tolerance of the pathogen by the immune system.

Vaginal microbiota may play a relevant role in the clearance or persistence of HPV [7]. In a previous paper, we demonstrated that bacterial species common in vaginal dysbiosis induce an unbalanced T cell differentiation that exacerbates inflammatory processes and may compromise viral clearance [16].

Chronic inflammation induced by bacterial species other than *Lactobacillus* is actually retained as the mechanism linking the vaginal environment to the neoplastic progression of HR-HPV-infected epithelial cells [17,18].

Chronic inflammatory status results in tissue damage, influx of neutrophils, and increased ROS production with consequent epithelial genomic instability, genotoxicity, and aberrant proliferation [19,20,21]. In accordance with this hypothesis, metabolomic profiles associated with cancers are also associated with cervical inflammation and HPV infections [22], and increased levels of proinflammatory cytokines (IL-1α, IL-1β, IL-6, IL-8, and TNF-α) have been shown in the vaginal fluids of women with cervical intraepithelial neoplasia [23].

In addition to ROS production, elevated levels of NF-kB activation also characterize chronic inflammation. The prolonged activation of NF-kB may play a role in the neoplastic transformation of the cells and its role in HR-HPV induced cancer has been extensively studied [24]. NF-kB is an important player in host response against infections since it induces the expression of inflammatory cytokines, chemokines, and their receptors. Beyond their role in immunity, all of these factors contribute to keeping the level of NF-kB activation high, and persistent inducing and amplifying side effects of NF-kB such as cyclin D1 and c-MYC transcription [25,26,27], which in turn, play important roles in cancer progression. Finally, NF-kB induces the production of host-deaminase as the APOBEC proteins and activation-induced cytidine deaminase (AID) enzymes, two families of proteins mainly involved in host-defense against viral infections that, however, can also mutate host DNA and contribute to cancer development [28].

The relevance of the E6 and E7 viral oncoproteins in the neoplastic transformation of cervical epithelial cells is beyond question. E6 and E7 directly affect important cell cycle checkpoints by inducing the degradation of p53 (tumor suppressor protein) and pRb (retinoblastoma protein), and the inhibition of cyclin dependent kinase (CDK) inhibitors (p21, p27, p16) [29]. E6 and E7 oncoproteins represent the major contributor to HPV-induced neoplastic initiation and the progression of carcinogenesis.

The co-expression of *E6* and *E7* usually constitutes the perfect environment for sustained proliferation. In fact, E7-mediated pRb disintegration induces cellular growth that could be stabilized by p53 in the case this protein is functional and is not degraded by the E6/E6AP complex.

Despite the relevance of E6 and E7 in HR-HPV induced neoplasia, less data are available on the direct or inflammatory-mediated effects of vaginal bacteria on *E6/E7* gene activation and/or protein production.

Soluble products of vaginal lactobacilli were reported as modulators of HPV-oncogene expression [30]; a positive association between the abundance of *Sneathia* and *Megasphaera* species and the expression of *E6* and *E7* genes was reported in cervical swabs from women with CIN-2/3 or cervical cancer [31].

In this study, we selected SiHa cells, a cell line with a few copies of HPV-16 integrated in the genome, as the experimental model of cervical neoplasia [32] and evaluated the effects of representative vaginal bacterial species on the *E6/E7* oncogene expression and the production of the relative oncoproteins. We compared the effects induced by vaginal dysbiosis bacteria (*G. vaginalis*, *A. vaginae*, *P. bivia*, *M. micronuciformis*) with those induced by vaginal lactobacilli.

Our data showed that all vaginal lactobacillus species downregulated the oncogene expression. *L. crispatus*, the most representative species of vaginal lactobacilli [2,7,31] significantly modulated the oncogene expression and the production of the relative oncoproteins, suggesting that, in addition to preventing HPV infection [7,9,33], this species might also control viral replication and genomic integration. *L. gasseri* behaves similarly to *L. crispatus*.

These data agreed with those obtained by Wang that used the culture supernatant of vaginal lactobacilli and CaSki cells (with ~600 copies of integrated HPV-16 genome) as the experimental model [30] and also suggest that different bacterial species are responsible for gene modulation in SiHa cells.

Despite the modulation of the *E7* oncogene induced by *L. iners*, we found that the E7 protein amount was increased compared to untreated cultures, suggesting that *L. iners* may activate pathways that increase the stability of the oncoprotein.

The co-expression of *E6* and *E7* constitutes the perfect environment for a sustained proliferation of the transformed cells [34]. However, since E6 and E7 may affect different pathways of cell cycle regulation and apoptosis, we cannot rule out that *L. iners* may play a role in HR-HPV-induced cancer. According to these data, a recent metanalysis suggests that *Lactobacillus crispatus*, but not *L. iners*, may be the specific protective factor against hrHPV infection, CIN, and CC [33]. Further studies are necessary to explore this point.

Among the vaginal dysbiosis bacteria, *G. vaginalis* and *M. micronuciformis* behave the opposite to *L. crispatus*: they increase the oncogene expression and the production of oncoproteins, suggesting a role for these species in viral-induced cancerogenesis. In contrast, *P. bivia* is a bacterial species able to significantly decrease the oncogene expression and the production of at least one oncoprotein (E7).

A recent survey showed a positive correlation among the abundance of *Prevotella* species in the vaginas of women with persistent HR-HPV and the expression of TLR4, NF-kB, C-myc, and hTERT by host cervical cells, suggesting that the overgrowth of this species may affect the rate of cervical lesions in women with persistent HPV infections [35].

As above reported, NF-kB plays an important role in HR-HPV induced cancer [24], since this pathway also leads to cyclin D1 and c-MYC transcription [25,26,27], which in turn also plays an important role in cancer progression.

The role of the TLR4 and NF-kB pathways in the expression of viral oncoproteins is still controversial.

It has been proposed that, at least in the initial phase of viral infection, the oncoproteins E6 and E7 inhibit the NF-kB activity [36], blocking an initial immune response that would be able to counteract the viral infection.

We have not studied the effects of vaginal bacteria on NF-kb activation because indirect evidence suggests that this pathway was not significantly activated by vaginal bacteria in SiHa cells. The production of the inflammatory cytokines IL-1β and TNFα and the chemokine IL-8 by SiHa cells is poorly stimulated by vaginal bacteria [16], suggesting a not significant activation of the NK-kB pathway in our experimental condition.

We are aware that these data do not authorize ruling out that vaginal bacteria affect the NF-kB dependent pathways differently. Further studies are needed to detail the molecular mechanisms activated by *L. crispatus* or by *Prevotella* species that lead to the inhibition of oncogene expression.

To obtain further evidence that *G. vaginalis* and *M. micronuciformis* interfere in the E6/E7-mediated oncogenic process, we investigated, in the same cultures, the production of p53 and pRb, the main targets of E6 and E7 oncoproteins, respectively.

E6 binds the cellular ubiquitin ligase E6AP (E6-associated protein) and induces the proteolytic degradation of p53 [37]. E7 preferentially binds the hypophosphorylated form of pRb and the complex E7/pRb functionally inactivates and degrades pRb, inducing, as a consequence, the release of E2 transcription factors and cyclin transcription, forcing the cells through premature S-phase entry [38,39,40].

We found a significant reduction in p53 and pRb oncosuppressors in cultures of SiHa cells with *M. micronuciformis*, in addition to an increased percentage of cells in the S phase of cell cycle after 24 h of culture. Thus, these data suggest that at least *M. micronuciformis* may accelerate neoplastic transformation by inducing the production of viral oncoproteins and the degradation of important cellular oncosuppressors.

Large prospective multicenter studies are needed to ascertain whether an abundance of *M. micronuciformis* with or without *G. vaginalis* can really affect the HR-HPV-induced neoplastic process.

## 4. Materials and Methods

### 4.1. Bacterial Strains

A collection of 8 vaginal bacterial reference strains were included in the study, and the related features are reported in Table 1.

*L. crispatus* (JV-V01), *L. gasseri* (SV-16A), and *L. iners* (UPII-60-B) were used as representative of CST-I, II, and III, respectively [41,42]. *G. vaginalis* was selected as representative of CST-IV and two strains isolated from women with bacterial vaginosis with Nugent Score 5 (49145/JCP-7276) or score 8 (14019/JCP-7659) were selected, respectively. *A. vaginae* (DSM-15829), *M. micronuciformis* (DNF00954), and *P. bivia* (DNF 00188) were also included as representative of CST-IV.

### 4.2. Bacterial Cultures

Bacteria were cultured on tryptic soy agar (TSA), composed of tryptic soy broth (Oxoid, Basingstoke, United Kingdom) and 1g/L of bacto-agar (Sigma Aldrich, St. Louis, MO, USA), supplemented with 5% defibrinated horse blood (Oxoid, Basingstoke, UK). Bacterial cultures were incubated at 36 ± 1 °C for 72 h in anaerobic conditions (AnaeroGen™, Thermo Fisher Scientific, Waltham, MA, USA) in a jar to create the ideal growth conditions (CO_2_: 9–13%; O_2_ < 0.01%).

Bacterial strains were then subcultured in liquid medium using TSB with 5% defibrinated horse blood (Oxoid, Basingstoke, United Kingdom). The bacterial concentration was measured using the DensiCHECK densitometer (bioMérieux, Marcy l’Étoile, France) after 1 mL of culture centrifugation at 4000× *g* for 5 min and pellet resuspension in 1 mL of physiological solution. The resulting McFarland value was used to calculate the bacterial concentration CFU/mL (colony forming units/mL) for the following standardization at a ratio of 1:50 with SiHa cells.

### 4.3. Epithelial Cell Culture

The SiHa cell line, isolated from squamous cell carcinoma and containing the HPV-16 genome (1 to 2 copies per cell), was obtained from ATCC^®^ (ATCC^®^ HTB35™). The SiHa cells were cultured in DMEM (Euroclone, Pero, Italy) supplemented with 10% FBS (fetal bovine serum), 1% L-glutamine, 1% penicillin and streptomycin (Euroclone, Pero, Italy) at 37 °C in the presence of 5% of CO_2_.

### 4.4. Gene Expression

RNA was extracted from 10^6^ cells using a Total RNA Extraction Kit (RBCBioscience, Real Genomics, New Taipei City, Taiwan cat. #YRB100) and quantified using a Nanodrop (Thermo Fisher, Waltham, MA, USA) and stored at −80 °C.

For each sample, we reverse-transcribed 2 ug of RNA using PrimeScript RT-PCR (TaKara, Kusatsu, Japan, cat. #RR047A). Real-time PCR was performed by using a QuantiNova SYBR Green PCR Kit (Qiagen, Hilden, Germany, cat. #208056) as described by Paccosi S et al. [41]. The 18S gene was used as the housekeeping gene. The primers used in this work are reported in Table 2.

### 4.5. Western Blot Analysis

A total of 1 × 10^6^ SiHa cells were cultured in DMEM for 6 h with and without live bacterial cells (50 MOI/cell).

Cells were lysed with RIPA buffer in the presence of a phosphatase/protease inhibitor cocktail (Sigma-Aldrich), centrifuged at 12,000 g, and 40 μg of proteins/lane was loaded onto Stain Free gel (Bio-Rad Hercules, CA, USA) SDS-PAGE and blotted onto PVDF filters and Millipore Immobilon Transfer membrane (Millipore, Sigma-Aldrich) as described [43].

Membranes were stained with mouse mAb anti-HPV16 E6/18 E6 (cat. #C1P5: sc-460, Santa Cruz Biotechnology Inc., Heidelberg, Germany), mouse mAb anti-HPV16 E7 (cat. #NM2:sc-65711, Santa Cruz Biotechnology Inc.), mouse mAb anti-p53 (D0-7) (cat. #48818, Cell Signaling Technology, Danvers, MA, USA), mouse mAb anti-pRb (cat. #9309, Cell Signaling Technology), rabbit anti-PpRb Ab (cat. #9308, Cell Signaling Technology), mouse anti-α -tubulin (B-5-1-2) (cat. #sc:-23948, Santa Cruz Biotechnology Inc.), rabbit anti-GAPDH antibodies (cat# sc:2118, Cell Signaling Technology) at 1:1000 final dilution. Goat anti-mouse IgG (H + L)-HRP conjugate or goat anti-rabbit IgG (H + L) (Human IgG Adsorbed) horseradish peroxidase conjugate (cat. #170-6516; cat. #170-6515, Bio-Rad, Hercules, CA, USA) were used as secondary antibodies at the final dilutions (1:2000). Reactions were visualized by the ECL detection system as recommended by the manufacturer (Bio-Rad, Hercules, CA, USA). The intensity of proteins of interest was acquired by the ChemiDocTouch System (Bio-Rad, Hercules, CA, USA). The densitometric analysis was expressed as the ratio between the interest protein and α-tubulin or the GAPDH housekeeping proteins by ImageLab software v.5.2 (Bio-Rad, Hercules, CA, USA) [43].

### 4.6. Cellular Cycle Analysis

A total of 10 × 10^4^ SiHa cells were cultured in complete DMEM for 24 h in the presence or absence of live bacterial cells (50 MOI/cell). The cell cycle phase distribution (propidium iodide staining) was determined as previously reported [44] using a FACS Canto (Beckton & Dickinson, San Josè, CA, USA).

### 4.7. Statistical Analysis

Viability data were analyzed by the analysis of variance (ANOVA) with the Bonferroni *P* value adjustment method for multiple comparisons from five different experiments. Gene expression (real-time PCR) and protein production (Western blot densitometry) were analyzed using the pairwise Wilcoxon test and Kruskal–Wallis test with and or without the Holm *p* value adjustment method for multiple comparisons. A *p* value < 0.05 was considered statistically significant. Statistical analysis was performed using R software version 4.2.2 [45].

## 5. Conclusions

Our data confirm that vaginal lactobacilli and *L. crispatus* are particularly able to modulate the oncogene expression in SiHa cells. *M. micronuciformis*, and to a lesser extent, *G. vaginalis*, increased the production of viral oncoproteins and reduced the cellular oncosuppressors p53 and pRb.

A more detailed analysis of bacterial strains isolated from the vaginal fluids of women with early cervical lesions (CIN-1) may help to identify the factor/s able to affect the oncogenic process induced by viral proteins.

## Figures and Tables

**Figure 1 ijms-24-07173-f001:**
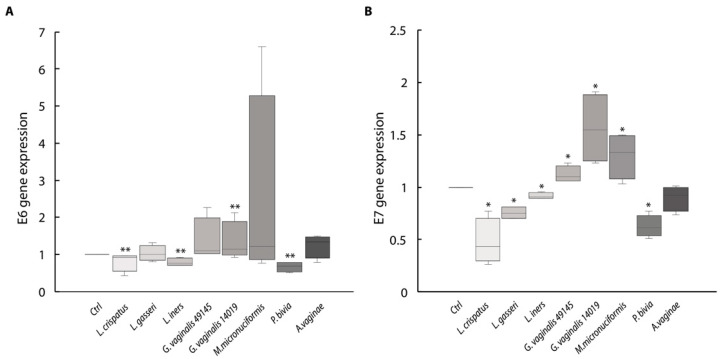
Effects of the bacterial strains on the expression of the HPV16 *E6* (**A**) and *E7* (**B**) oncogenes in the SiHa cells. Data are expressed as fold-change in the mRNA relative amount with respect to the untreated cultures (Ctrl). The box plot shows the median f five different experiments and the whisker was calculated using the formula IQR × 1.5. Statistical analysis was performed by the Kruskal–Wallis test with Holm–Bonferroni *p* value adjustment. *p* value *E6* = 0.33; *p* value * *E7* = 0.001; ** *p* value vs. Ctrl 0.005.

**Figure 2 ijms-24-07173-f002:**
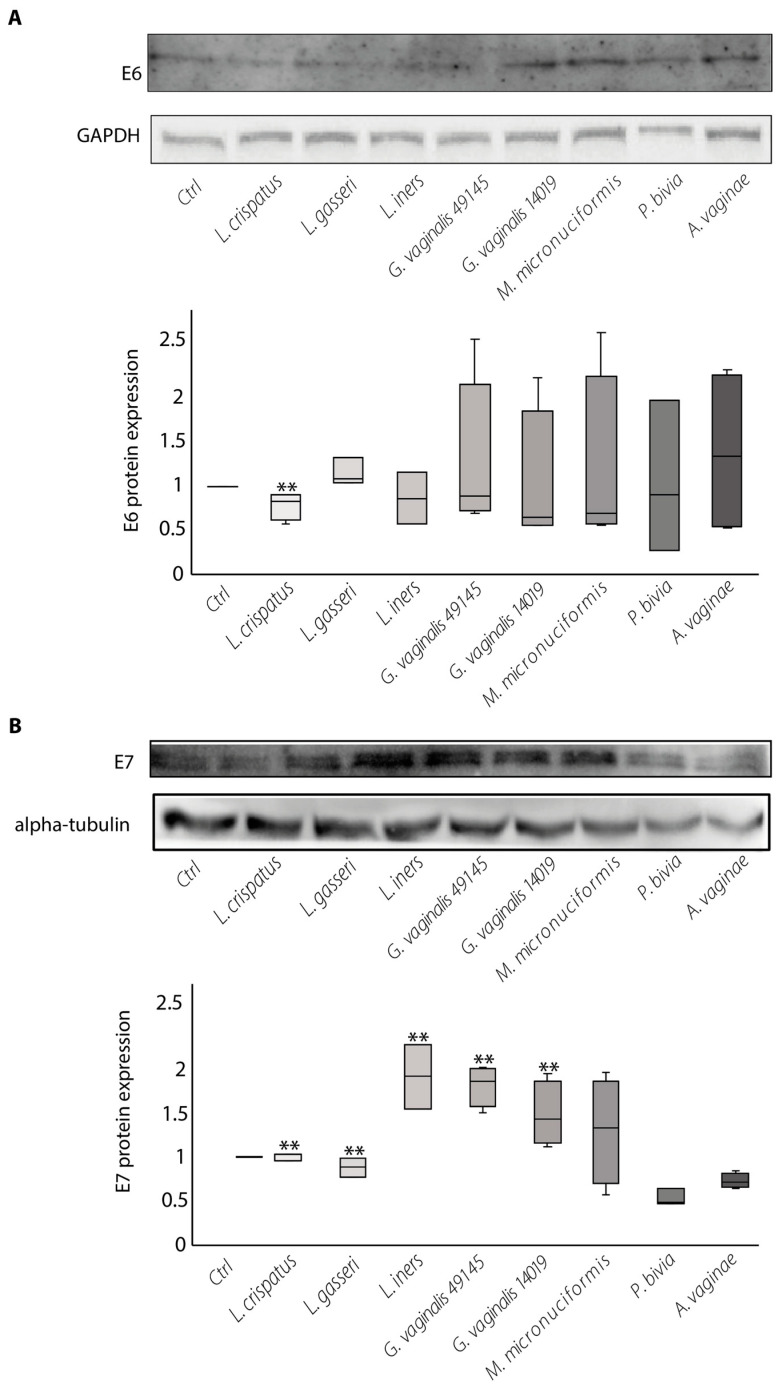
The effects of bacterial strains on the production of HPV-16 E6 (**A**) and E7 (**B**) oncoproteins in the SiHa cells. The amount of E6 and E7 proteins were evaluated by Western blot analysis. The figure shows the results of one representative experiment out of five performed. The box plot below represents the median of five different experiments and the whisker was calculated using the formula IQR × 1.5. Statistical analysis was performed by the Kruskal–Wallis test and pairwise Wilcoxon test comparisons. *p* value E6 = 0.7; *p* value E7 = 0.002; ** *p*-value ≤ 0.005 of bacterial stimulated cultures vs. untreated cell cultures (Ctrl).

**Figure 3 ijms-24-07173-f003:**
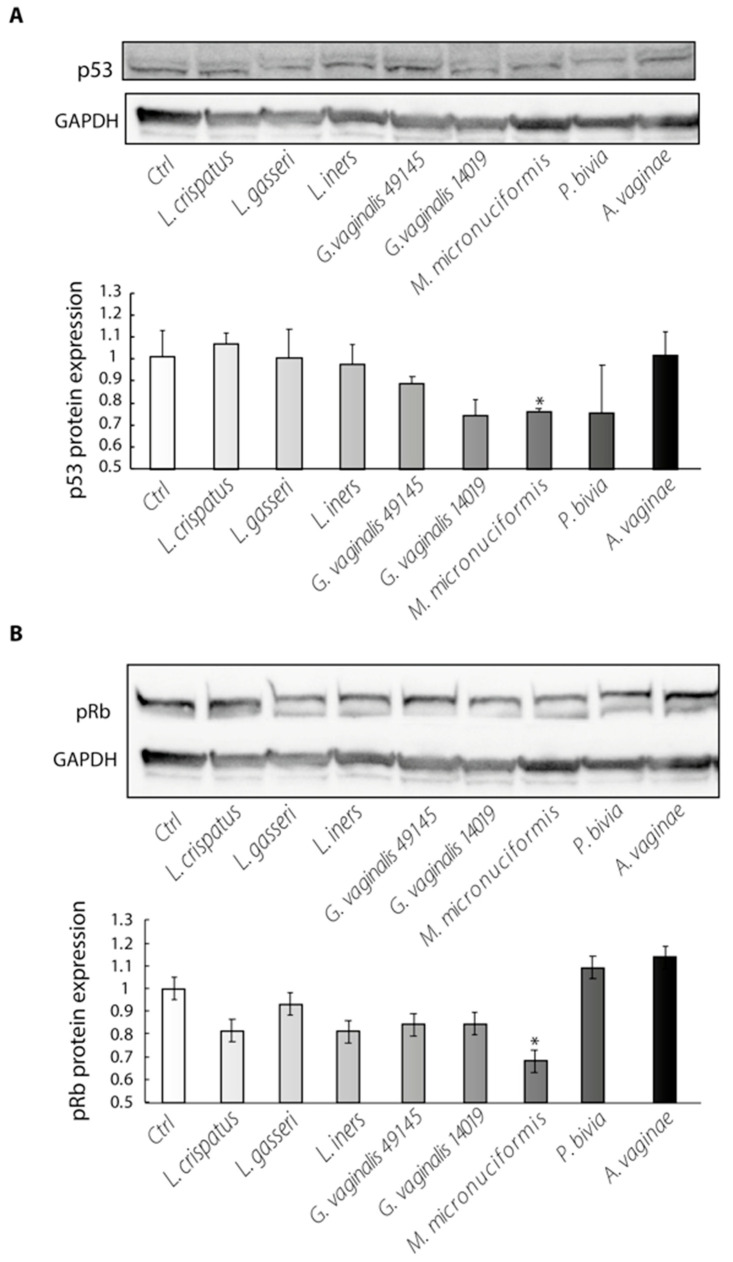
The effect of bacterial strains on the amount of p53 (**A**) and pRb (**B**) proteins in SiHa cells. Western blot analysis with anti p53 and anti P-pRb antibodies. Data from one representative experiment out of five performed are shown. Bar graph shows the mean of five different experiments ± SE. Statistical analysis was performed by the Kruskal–Wallis test with Holm–Bonferroni *p* value adjustment. * *p*-value ≤ 0.05 of bacterial stimulated cultures vs. untreated cell cultures (Ctrl).

**Figure 4 ijms-24-07173-f004:**
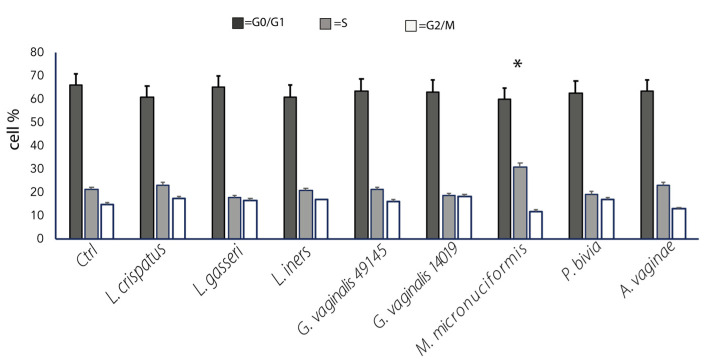
The effect of bacterial strains on the SiHa cell cycle. Results are shown as the mean percentage of cells ± SE in each phase of the cell cycle (G0/G1, S, and G2/M) recorded by FACS analysis (CSFE fluorescent staining). The bar graph represents the means ± SE of three different experiments. Statistical analysis was performed by the Student *t*-test. * *p*-value ≤ 0.05 of the bacterial stimulated cultures vs. untreated cell cultures (Ctrl).

**Table 1 ijms-24-07173-t001:** Reference bacterial strains.

CST	Family and Genus	Species	Strain
I	*Lactobacillaceae*, *Lactobacillus*	*L. crispatus*	JV-V01
II	*Lactobacillaceae*, *Lactobacillus*	*L. gasseri*	SV-16A
III	*Lactobacillaceae*, *Lactobacillus*	*L. iners*	UPII-60-B
IV	*Bifidobacteriaceae*, *Gardnerella*	*G. vaginalis*	49145/JCP-7276
IV	*Bifidobacteriaceae*, *Gardnerella*	*G. vaginalis*	14019/JCP-7659
IV	*Veillonellaceae*, *Megasphaera*	*M. micronuciformis*	DNF-00954
IV	*Prevotellaceae*, *Prevotella*	*P. bivia*	DNF-0018
IV	*Atopobiaceae*, *Atopobium*	*A. vaginalis*	DSM-15829

**Table 2 ijms-24-07173-t002:** Oligonucleotide sequences used in this study.

Gene	Forward 5′-3′	Reverse 3′-5′
*18S*	ATTAAGGGTGTGGGCCGAAG	GGTGATCACACGTTCCACCT
*E6*	CGACCCAGAAAGTTACCA	AGCAAAGTCATATACCTCACG
*E7*	GCCACCATGCATGGAGATACACCTACA	GATCAGCCATGGTACATTATGG

## Data Availability

Not applicable.

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
