# Peer review of "Bacterial Species from Vaginal Microbiota Differently Affect the Production of the E6 and E7 Oncoproteins and of p53 and p-Rb Oncosuppressors in HPV16-Infected Cells"

_ijms, 2023, doi:10.3390/ijms24087173_

Round 1

Reviewer 1 Report

The study addressed by the authors is interesting and could provide a contribution to the understanding of how the vaginal microbiota influences the progression of HPV-associated lesions towards cancer. However, in its present form, the paper presents many critical points, both formal and substantial. The main criticality is represented by the fact that the data are not properly supported by the figures. Below, I point out problems and related suggestions.

Major points

-The authors use HPV16-positive SiHa cells for their studies. Since there are no alternative in vitro models of HPV infection, the choice is acceptable, but SiHa are immortalized cells isolated from a patient's squamous cell carcinoma, and as such cannot be considered as an experimental model of CIN 1, which is a low-grade epithelial lesion.

-The major point is that the data presented do not adequately support the conclusions.

Fig.1. The authors claim that the WB shown is one representative of 5 performed. Nevertheless, the statement that “L. crispatus significantly decreased the basal expression of both E6 and E7 proteins” does not match the figure, and one just has to rely on the plots. More representative WBs would be preferable. Furthermore, RT PCR results should be shown to support data on increase and decrease of expression.

Page 3, lanes 95-96: from the WB in fig. 1, not only G. vaginalis strains and M. micronuciformis seem to increase the E7 expression, but also L. iners (this is also indicated by asterisks). Furthermore, it can be deduced that some asterisks for significance are skipped.

And what about P. bivia and A. vaginae? Again, the figures (both WB and plots below) do not reflect what is stated in the text.

Page 4, lanes 101-103: the statement that L. crispatus and L. gasseri are protective species is overstating, as is the claim that G. vaginalis and M. micronuciformis facilitate neoplastic progression. At most, a contribution to decreasing or increasing the risk of tumor progression by virtue of the effect on oncoproteins could be hypothesized.

 Page 4, lanes 115-116: again, from Fig. 2 it does not seem that M. micronuciformis is the only strain to affect p53 and pRb expression (see the bar relative to G. vaginalis 14019 for p53, and at least other 4 bars for pRb).

Minor points

-Introduction. Ref 12 should refer more specifically to the percentage of cervical cancer caused by high-risk HPVs worldwide while it refers to the E7/pRb binding.

Results, title and throughout the text: “expression” is preferrable to “synthesis” or to “production” when referring to the E6 and E7 proteins in cells.

All numbers in the y-axis of the graphs are in the Italian decimal system.

The term "untreated" would be preferable to "unstimulated" for controls.

Page 7, lane 192: what do the authors mean for “simultaneously”? E6 and E7 are able to affect different pathways of proliferation and apoptosis, and it is not mandatory to block both proteins to alter such pathways.

Page 8, Table 1: bacterial strains should be listed in the same order as in WBs and plots.

Materials and Methods. Page 8, lane 253: the antibody used is Phospho-Rb (Ser807/811), detecting the phosphorylated pRb. This should be specified and could probably account for the apparently different MW of the bands in Fig. 2.

In general: I noticed that uncropped WB images are better than images in the figures, and the pRb images do not even show double bands. The p53 images do not even seem the same as in the manuscript.

In Fig. 2, panels for p53 and pRb could be named as A and B.

In Conclusions, page 9, lanes 287-288: even if said doubtfully and even if the possibility has been suggested in Ref #10, results are overstated. In this case, differences are too small to be able to represent a diagnostic marker of tumor progression, much less to guide a therapeutic decision.

Author Response

Referees’ comments

The study addressed by the authors is interesting and could provide a contribution to the understanding of how the vaginal microbiota influences the progression of HPV-associated lesions towards cancer. However, in its present form, the paper presents many critical points, both formal and substantial. The main criticality is represented by the fact that the data are not properly supported by the figures. Below, I point out problems and related suggestions.

Major points

  • The authors use HPV16-positive SiHa cells for their studies. Since there are no alternative in vitro models of HPV infection, the choice is acceptable, but SiHa are immortalized cells isolated from a patient's squamous cell carcinoma, and as such cannot be considered as an experimental model of CIN 1, which is a low-grade epithelial lesion.

Author’s response: We accepted the criticism and corrected the description of SiHa cells as follows:

Lines 81-83: SiHa cells, harboring few copies of integrated copies of HPV-16, were used as experimental model of HR-HPV transformed cells and co-cultured with live bacterial cells for 6 hours.

  • The major point is that the data presented do not adequately support the conclusions. Fig.1. The authors claim that the WB shown is one representative of 5 performed. Nevertheless, the statement that “ crispatus significantly decreased the basal expression of both E6 and E7 proteins” does not match the figure, and one just has to rely on the plots. More representative WBs would be preferable.

Author’s response: We accepted the criticism. We loaded a new figure (Figure 2a) showing western blot analysis with anti-E6 antibody, in which we observe more evident differences in protein amount among bacterial-stimulated and untreated cultures. We would prefer to maintain the figure representing changes in E7 protein (Figure 2b).

  • Furthermore, RT-PCR results should be shown to support data on increase and decrease of expression.

Author’s response: In lines 77-104 we added data from RT-PCR for HPV-E6 and E7 genes as suggested (Figure 1, panel A and B).

  • Page 3, lanes 95-96: from the WB in fig. 1, not only vaginalis strains and M. micronuciformis seem to increase the E7 expression, but also L. iners (this is also indicated by asterisks). Furthermore, it can be deduced that some asterisks for significance are skipped.

And what about P. bivia and A. vaginae? Again, the figures (both WB and plots below) do not reflect what is stated in the text.

Author’s response: We thank the reviewer for the criticism. Data obtained by culturing SiHa cells with L. iners, P. bivia and A. vaginae are now described in the “Results” section and commented in the discussion section (Lines 84-96 and 255-259).

Page 4, lines 101-103: the statement that L. crispatus and L. gasseri are protective species is overstating, as is the claim that G. vaginalis and M. micronuciformis facilitate neoplastic progression. At most, a contribution to decreasing or increasing the risk of tumor progression by virtue of the effect on oncoproteins could be hypothesized.

Author’s response: We thank the reviewer for the criticism. We corrected the sentence on the risk of tumor progression as follow:

Lines 116-118: Overall, these data indicate L. crispatus and L. gasseri as vaginal Lactobacillus species that are able to modulate the expression of E6 and E7 oncogenes and the production of the relative oncoproteins.

  • Page 4, lanes 115-116: again, from Fig. 2 it does not seem that micronuciformis is the only strain to affect p53 and pRb expression (see the bar relative to G. vaginalis 14019 for p53, and at least other 4 bars for pRb).

Author’s response: It is true.  We stressed the results obtained with M. micronuciformis because statistical analysis showed significant differences between untreated cultures and M. micronuciformis treated cultures (Lines 141-144). Moreover, percentage of cells in S phase was significantly increased only in cultures with M. micronuciformis.

Minor points

  • Ref 12 should refer more specifically to the percentage of cervical cancer caused by high-risk HPVs worldwide while it refers to the E7/pRb binding.

Author’s response: Ref n. 12, which reports the interaction between E7 and pRb, is mentioned in the results section as reported:

Lines 135-136: The HPV oncoproteins E6 and E7 target to ubiquitination p53 and pRb proteins respectively [11,12].

Results, title and throughout the text: “expression” is preferrable to “synthesis” or to “production” when referring to the E6 and E7 proteins in cells.

Author’s response: We accept the criticism, and we changed the term “expression” to “production”, referring to proteins.

  • All numbers in the y-axis of the graphs are in the Italian decimal system.

Author’s response: Sorry for the typo. All numbers in the y-axis of the graphs have been changed in the international decimal system.  

  • The term "untreated" would be preferable to "unstimulated" for controls.

Author’s response: We accepted the criticism, and we changed the term “unstimulated” to “untreated” everywhere in the text.

  • Page 7, lane 192: what do the authors mean for “simultaneously”? E6 and E7 are able to affect different pathways of proliferation and apoptosis, and it is not mandatory to block both proteins to alter such pathways.

Author’s response: It is correct that E6 and E7 affect different pathways of proliferation and apoptosis. However, it is reported that the joint action of HPV E6 and E7 oncoproteins is necessary for HPV-induced cancer. However, we accepted that the term “simultaneously” is not correct and changed the period as follows:

Lines 218-220: The co-expression of E6 and E7 usually constitutes the perfect environment for sustained proliferation. In fact, E7-mediated pRb disintegration induces cellular growth that could be stabilized by p53 in the case this protein is functional and not degraded by the E6/E6AP complex.

  • Page 8, Table 1: bacterial strains should be listed in the same order as in WBs and plots.

Author’s response: Revised as suggested, the order of the list in Table 1 has been changed.

  • Materials and Methods. Page 8, lane 253: the antibody used is Phospho-Rb (Ser807/811), detecting the phosphorylated pRb. This should be specified and could probably account for the apparently different MW of the bands in Fig. 2

Author’s response: We thank the referee for the criticism. The legend now reports the use of anti-PpRb (Line 147).

  • In general: I noticed that uncropped WB images are better than images in the figures, and the pRb images do not even show double bands. The p53 images do not even seem the same as in the manuscript.

Author’s response: In the uncropped p53 image it is represented the same western blot than figure 3 (ex-Figure 2). The only difference is that we loaded 10 well (the last with lysates of Enterococcus faecalis stimulated cultures. It is possible that this generates confusion.

  • In Fig. 2, panels for p53 and pRb could be named as A and B.

Author’s response: Thank you for the suggestion. Now Figure 3 (ex-Figure 2) has been divided in panel A and panel B as suggested.

  • In Conclusions, page 9, lanes 287-288: even if said doubtfully and even if the possibility has been suggested in Ref #10, results are overstated. In this case, differences are too small to be able to represent a diagnostic marker of tumor progression, much less to guide a therapeutic decision.

Author’s response: We accept this criticism. The last period of Conclusion section has been changed as follows:

Lines 382-384: A more detailed analysis of bacterial strains isolated from vaginal fluids of women with early cervical lesions (CIN-1) may help to identify the factor/s able to affect the oncogenic process induced by viral proteins.

Reviewer 2 Report

In this work, the authors have elucidated the role of various bacterial species that constitute the vaginal microbiota on curbing the progression of cervical cancer in invitro model using HR-HRV cervical epithelial cells, SiHa cells. The authors measured the various protein levels that were previously shown to prevent cervical cancer level including p53, pRB along with the levels of 2 viral oncoprotein E6 and E7. Interestingly, the authors found that treating with G. vaginalis lead to 2.5-fold increase in the level of viral oncoprotein E6 and treatment with Lactobacillus iners and G. vaganilis leads to increased E7 oncoprotein. Furthermore, the authors found that treatment with M. micronuciformis lead to decrease in p53 and pRb levels. Finally, the authors have tracked down the progression through the cell cycle as loss or decrease in the level of p53 and pRb can lead the epithelial cells to enter the S-phase. The authors found that treatment with M. micronuciformis lead to significant increase in the number of cells in S-phase. This was an expected outcome as treatment with this bug leads to decrease in the p53 level.

Overall, the work is interesting as the authors have found out the preliminary evidence about the contribution of various bacterial species in the cervical cancer. One thing that came forward from their study is some of the bacterial species can speed the progression of the cervical cancer by decreasing the levels of the p53.

I’m outlying my comment below:

1.Although the study has outlined an important aspect of the cervical cancer, the data is preliminary as no molecular mechanism of how this phenotype has been explained. Further, this is an invitro model.

2. The authors should repeat this in the murine model of infection to validate the findings.

3. The discussion section needs much work. The authors should elaborate about why they think certain bacterial species is showing a phenotype whereas other are neutral.

4. Figure 2” There is a gap in the both the panels of Figure 2 between L. iners and G. vaginalis 49145. Is this a formatting error. The authors should reformat it.

Author Response

  • Although the study has outlined an important aspect of the cervical cancer, the data is preliminary as no molecular mechanism of how this phenotype has been explained.

Author’s response: We thank the reviewer for the criticism. We are aware that these data are preliminary. However, they may help to understand why in many clinical studies the taxa studied are found associated with HSIL and cervical cancer.

All experiments reported in the paper were performed with live bacterial cells.

We are currently performing experiments with bacterial lysates and/or purified toxins to study selected molecular pathways (MAPK, NF-kB, etc.).

  • Further, this is an in vitro The authors should repeat this in the murine model of infection to validate the findings.

Author’s response: We absolutely agree with the referee. In vivo experiments are beyond the scope of this work and will be the subject of further investigation.

  • The discussion section needs much work. The authors should elaborate about why they think certain bacterial species is showing a phenotype whereas other are neutral.

Author’s response: The discussion section has been expanded as suggested (Lines 173-296).

  • Figure 2” There is a gap in the both the panels of Figure 2 between L. iners and G. vaginalis 49145. Is this a formatting error. The authors should reformat it.

Author’s response: Corrected as suggested.

Round 2

Reviewer 1 Report

I thank the authors for accepting most suggestions. Further minor remarks:

Actually, the new WB in Fig. 2A is qualitatively worse than the previous one although the decrease of E6 expression is more evident. 

I meant that "expression" is preferrable to "synthesis" and also to "production". Instead, the authors put "production" everywhere, that was not my suggestion. Please feel free to use the terms you like.

The legend of figure 1 reports typing errors of the decimal system.

Author Response

Actually, the new WB in Fig. 2A is qualitatively worse than the previous one although the decrease of E6 expression is more evident. 

Authors response:

Sorry for the low quality of Fig. 2A, we have now included in the manuscript an image presenting a higher quality.

I meant that "expression" is preferrable to "synthesis" and also to "production". Instead, the authors put "production" everywhere, that was not my suggestion. Please feel free to use the terms you like.

Authors response:

Sorry for the confusion. In order to be coherent in the text and title, we have now used for E6, E7 oncogenes the term expression, and for E6, E7, p53 and pRB proteins the term production.

The legend of figure 1 reports typing errors of the decimal system.

Authors response:

Corrected as suggested.

Reviewer 2 Report

The authors have improved the manuscript and made the corrections in the discussion section.

Author Response

Thanks for your comments and suggestions.